# The Emerging Roles of Long Non-Coding RNAs in Intellectual Disability and Related Neurodevelopmental Disorders

**DOI:** 10.3390/ijms23116118

**Published:** 2022-05-30

**Authors:** Carla Liaci, Lucia Prandi, Lisa Pavinato, Alfredo Brusco, Mara Maldotti, Ivan Molineris, Salvatore Oliviero, Giorgio R. Merlo

**Affiliations:** 1Department of Molecular Biotechnology and Health Sciences, University of Torino, 10126 Torino, Italy; carla.liaci@unito.it (C.L.); lucia.prandi@edu.unito.it (L.P.); 2Department of Medical Sciences, University of Torino, 10126 Torino, Italy; lisa.pavinato@unito.it (L.P.); alfredo.brusco@unito.it (A.B.); 3Department of Life Sciences and System Biology, University of Torino, 10124 Torino, Italy; mara.maldotti@unito.it (M.M.); ivan.molineris@unito.it (I.M.); salvatore.oliviero@unito.it (S.O.)

**Keywords:** lncRNAs, intellectual disability, neurodevelopmental disorders, gene networks, neuronal differentiation, systems biology

## Abstract

In the human brain, long non-coding RNAs (lncRNAs) are widely expressed in an exquisitely temporally and spatially regulated manner, thus suggesting their contribution to normal brain development and their probable involvement in the molecular pathology of neurodevelopmental disorders (NDD). Bypassing the classic protein-centric conception of disease mechanisms, some studies have been conducted to identify and characterize the putative roles of non-coding sequences in the genetic pathogenesis and diagnosis of complex diseases. However, their involvement in NDD, and more specifically in intellectual disability (ID), is still poorly documented and only a few genomic alterations affecting the lncRNAs function and/or expression have been causally linked to the disease endophenotype. Considering that a significant fraction of patients still lacks a genetic or molecular explanation, we expect that a deeper investigation of the non-coding genome will unravel novel pathogenic mechanisms, opening new translational opportunities. Here, we present evidence of the possible involvement of many lncRNAs in the etiology of different forms of ID and NDD, grouping the candidate disease-genes in the most frequently affected cellular processes in which ID-risk genes were previously collected. We also illustrate new approaches for the identification and prioritization of NDD-risk lncRNAs, together with the current strategies to exploit them in diagnosis.

## 1. Introduction

A large fraction of the human genome is transcribed into non-coding RNAs, of which a good portion is represented by long non-coding RNAs (lncRNAs), namely untranslated RNAs composed of 200 or more nucleotides. They are transcribed by RNA polymerase II, contain a 5′mtG cap and a 3′ poly(A) tail and can be spliced, but lack the protein-coding function [1,2,3,4]. Thousands of lncRNAs have been annotated in diverse cell types and tissues, with specific patterns of spatio-temporal expression [5,6]. However, until recently, they have been largely overlooked and dismissed. Now, the non-coding genome is emerging as a major source of human diversity and has been found to participate in physiological and pathological processes, but its dynamics of expression, mechanisms of action, and functional roles remain poorly characterized [7]. At the molecular level, the most studied function of lncRNAs is the modulation (induction or repression) of gene expression, which includes the regulation of neighboring genes in cis [8], the complex formation with RNA binding proteins and chromatin modifiers [9,10], and the binding to specific genomic regions via the triple-helix formation in trans [11]. Still, they also work at the post-transcriptional level by competing with miRNAs [12], acting as splicing regulators, or promoting RNA degradation [13,14,15].

lncRNAs show a transcriptional kinetics not dissimilar to that of protein-coding genes, although with a lower frequency of transcriptional bursts [16]. Current knowledge indicates that the mechanism of the transcription and processing of most lncRNAs should be similar to those of protein-coding genes [17,18], in particular, distinct studies have suggested that lncRNA promoters share transcription factor binding sites with protein-coding genes [19,20,21]. lncRNAs are also expected to share epigenetic regulations with them including DNA-methylation and histone modifications. Most lncRNAs are transcribed from promoters with low CpG dinucleotide content, not marked by H3K4me3 or H3K27me3, features that correlate with their low expression level [17,22]. Histone deacetylation may be involved in lncRNA silencing, as in the case of the lncRNA LET [23]. miRNAs also serve as modulators of lncRNA promoter methylation [17]. For instance, miRNA-29 was reported to upregulate the expression of the lncRNA MEG3 by inhibiting DNA methyltransferase from methylating the promoter [24]. Another example is provided by miR-141, which was shown to promote dopaminergic neuronal differentiation via suppressing lncRNA-HOTAIR at the promoter level [25].

The expression of lncRNAs appears to be remarkably tissue-specific and somehow related to the expression of the neighboring genes compared to that of coding genes [5]. In particular, the physical proximity and the co-expression of a major fraction of them with developmental genes suggest their putative relevance in the organogenesis process. Their developmental dynamics of expression were recently analyzed in seven different species organs, showing signatures of functional enrichment [26]. However, they are rarely evolutionary conserved in different species, considering that only 12% of mouse lncRNAs have a human orthologue [5,27,28].

The brain is the organ with the greatest number of unique cell-type-specific lncRNAs [29,30]. In humans, it was estimated that 40% of the lncRNAs (approximately 4000–20,000 genes) are expressed only in the brain [31]. Some of them are selectively transcribed in vitro upon specific neuronal commitment or upon oligodendrocyte progenitor specification [32]. Because of their spatio-temporal expression specificity, a role for lncRNAs in brain development has been proposed and supported by studies in which the ablation of single lncRNA loci was shown to perturb mouse neural development [33,34]; lncRNAs have also been implicated in the REST and REST corepressor functions, controlling and being controlled, respectively, by the transcriptional regulators involved in neuronal and glial fate specification [35].

lncRNAs have also been hypothesized to be among the drivers of the evolution of the human brain. Many lncRNAs were positively selected by being conserved in their sequences in the human genome compared to that of other primates [7]. This positive selection appears to be even more relevant when compared to the selection experienced by the protein-coding genes related to the nervous system, which is surprisingly much lower [36,37]. Among the lncRNA loci that underwent human-specific positive selection, *HARF1* has been suggested to play a prominent role in driving cortical development. *HARF1* is highly expressed in a molecular defined neuronal subclass, the Cajal–Retzius neurons, which populate the cortical marginal zone from the early stages of development and guide the cortical lamination by the release of Reelin [38].

Despite the emerging importance of lncRNAs in the human brain, little is known about the mutations in their genomic loci or how they indirectly affect their expression and functionality. Genetic studies have examined lncRNAs as potential factors in complex human brain disorders; in particular, they have been implicated in neurodegenerative processes including Alzheimer’s [39] and Huntington’s disease [40]. However, evidence indicating that some lncRNAs may participate—or even cause—ID and other forms of NDD is accumulating.

ID is characterized by limited intellectual functioning and adaptive behavior that appear before the age of 18 [AIDD American Association 2010]. It is usually defined by an intelligence quotient <70 and severe deficiency in the environment and social milieu adaptation [41], which can be caused by genetic, prenatal, and environmental factors, representing a health issue in both the developed and developing countries [42]. ID can be classified by severity into four categories (i.e., mild, moderate, severe, profound) and it is divided into syndromic, which includes patients with one or more clinical co-morbidities in addition to ID, and non-syndromic, in which ID is the exclusive clinical feature [43].

The non-genetic causes or con-causes of ID are being heavily investigated [44,45,46,47]; concerning genetics, research has so far focused on the identification of mutated protein-coding genes inherited in a Mendelian fashion, mainly due to the availability of the exome sequencing derived data from ID patients and trios [48,49,50,51]. Such continuing effort has led to the identification of more than 200 confirmed ID-causative genes. Subsequent meta-analyses of these led to the recognition of a limited number of intracellular core regulatory processes on which the mutated genes converge [51]. However, a large fraction of ID and related NDD patients remains undiagnosed after exome-sequencing, opening the possibility of alterations in the non-coding genome. 

In this review, we aim to discuss the relevance of some lncRNAs in the pathoetiology of NDD, and specifically of ID, describing the in vitro and in silico approaches that could provide preliminary evidence and discuss the possible outcomes in the diagnoses of rare lncRNA-associated NDD.

## 2. lncRNAs from Stem and Progenitor Cells to Neurons

ID and related NDD are associated with defective neural differentiation and circuit formation [52], although the exact cellular phenotype underlying the distinct ID forms remains largely elusive, especially when considering that most of our knowledge derives from non-human models. While protein-coding genes have been extensively studied in the context of cell differentiation and circuit formation, the role of lncRNAs is just emerging.

lncRNAs have been shown to play key roles in distinct neurodevelopmental stages: in the neurogenic or gliogenic commitment of the neural stem cells (NCSs) and during the post-mitotic cell maturation stage, when neuronal or glial cells acquire their functional features [53,54].

Here, we subdivided the neurodevelopmental relevant lncRNAs in three groups, based on their broad function (transcriptional and post-transcriptional regulators or small peptide-coding) and intracellular location (nuclear and/or cytoplasmic).

### 2.1. lncRNAs Acting as Transcriptional Regulators 

The mechanisms of lncRNA-mediated transcriptional regulation include nuclear- or chromatin-enriched lncRNAs, which can essentially recruit transcriptional regulatory factors to a gene in cis or in trans, or influence the genome organization and chromatin architecture [55,56].

Two lncRNAs named the *Sox1 overlapping transcript* (***Sox1ot***) and *Sox2 overlapping transcript* (***Sox2ot***), both detected in the nucleus, are evolutionarily conserved and transcriptionally overlap with the *Sox1* and *Sox2* transcription factor genes [57,58,59,60], known to be involved in the maintenance of the stemness of pluripotent cells and NSCs [59,60,61]. 

*SOX1ot* is highly expressed during neural differentiation and acts as a decoy, interacting with HDAC10 to promote a SOX1-dependent regulatory mechanism of human brain development [59,62]. The other overlapping transcript, *SOX2ot*, together with *Paupar* and *rhabdomyosarcoma 2-associated transcript* (*RMST*), have been shown to play a role in neural development by regulating SOX2 expression, which in turn is required to attain the neural lineage commitment stage [61]. Specifically, *SOX2ot* has been shown to be expressed in the cerebral cortex, where it negatively regulates the self-renewal of NSCs physically interacting with the transcriptional regulator YY1. YY1 binds to the CpG island in the Sox2 locus to suppress the expression of Sox2 [60].

The lncRNA named ***RMST*** appears to be essential for neurogenesis, being implicated in the regulation of more than 1000 genes [63]. *RMST* promotes SOX2 expression and these together co-activate several neurogenic genes such as *Ascl1*, *NeurogG2*, *Hey2*, and *Dlx1*. *RMST* was found to promote the expression of SP8 and HEY1. It is transcriptionally repressed by REST and interacts with the RNA-binding proteins hnRNPA2/B1, which regulates splicing in neural progenitors [63].

The *Pax6 Upstream Antisense RNA* (***Paupar***) is divergently transcribed from the *Pax6* locus and participates in the regulation of neural differentiation and olfactory bulb neurogenesis via binding with the local genes *Pax6* and *Kap1* in a cis-acting manner. *Paupar* also acts in trans to control the neural gene expression on a large scale. In particular, knockdown of *Paupar* induces neurite outgrowth and neural differentiation [64]. *Paupar* physically interacts with the transcription factors PAX6 and KAP1, together with multiple regulatory regions (e.g., *2f2*, *E2f7*, *Cdc6*, *Cdkn2c*, *Kdm7a*, *Sox1*, *Sox2*, *Hoxa1*, *Hes1*) modulating the H3K9me3 deposition at the PAX6-bound sequences [65].

The lncRNA ***lncKdm2b*** facilitates neuronal differentiation during the early neurogenesis of cortical projection neurons. *lncKdm2b* allows for the expression of the coding gene *Kdm2b* in cis, promoting a permissive chromatin environment at the promoter level via binding with hnRNPAB [66].

The long nucleolus-specific lncRNA (***LoNA***) and its human orthologous RP11-517C16.2 harbor nucleolin-binding sites and specifically interact with NCL to suppress rRNA biosynthesis, influencing neuronal plasticity [67].

***lnc-NR2F1*** is a conserved chromatin-associated lncRNA that acts in trans, interacting with genomic regions to regulate neuronal cell maturation; it has been found to be mutated in human patients with autism spectrum disorder (ASD) and ID [68].

Other lncRNAs are important for synapse formation and transmission. The cloud-forming *Dlx5/Dlx6* ultraconserved enhancer lncRNA, ***Evf2***, is a regulator of the development of GABAergic interneurons, important for their synaptic transmission. *Evf2* forms a complex with DLX1 ribonucleoprotein and directly inhibits BRG1 ATPase and chromatin remodeling activities in the developing forebrain [69,70]. The transcriptional control exerted by *Evf2* occurs across long distances in the genome to regulate interneuron diversity [71]. Interestingly, it was also shown to contribute to the susceptibility to seizure in adults [71].

The lncRNAs ***NEAT1***, and *Metastasis-Associated Lung Adenocarcinoma Transcript1* (***MALAT1*/*NEAT2***) localizes to nuclear para-speckles and speckles, which contain various pre-mRNA splicing regulators. *NEAT1* is expressed in NSCs and is involved in oligodendrocyte differentiation [32,72]. *MALAT1* influences the synapse formation of cultured hippocampal neurons by altering the recruitment of SR family pre-mRNA-splicing factors [73]. *MALAT1* is also required for neurite outgrowth as its knockdown halts this process and promotes cell death via the suppression of Mitogen-Activated Protein Kinase (MAPK) signaling [74].

### 2.2. lncRNAs Acting as Post-Transcriptional Regulators

Some lncRNAs function by sequestering specific microRNA (miRNA) via base-pairing—an activity known as competing endogenous RNA (ceRNA)—and thereby indirectly regulating gene expression in the cytoplasm [75,76,77]. miRNAs are short non-coding RNAs (approximately 22 nucleotides long), suppressing mRNA translation and/or stability, which participate in all stages of neural differentiation during brain development [78].

Several lncRNAs interacting with miR-30e-3p, miR-431, and miR-147 were identified by microarray analysis in the hippocampus. Among these, ***Gm21284*** was characterized to function as a ceRNA and enhance the proportion of CHAT-positive cells during NSC differentiation [79]. 

***lncRNA*-*1604*** knockdown significantly suppresses neural differentiation; it acts as a ceRNA of miR-200c to regulate the key transcription factors zinc finger E-box binding homeobox 1/2 (ZEB1/2) [75]. 

The lncRNA ***Ube3a1*** acts as a ceRNA, inhibiting the activity of the plasticity-regulating miR-134 and other miR379–410 members, with an implication in Angelman syndrome and ASD [80].

lncRNA transcripts could generate several variants to execute functions in neurogenesis. The lncRNA C130071C03 Riken variants ***Rik*-*201*** and ***Rik*-*203*** have been proposed as modulators in the developing brain since they are activated by neurogenesis related transcript protein CCAAT/enhancer-binding protein β (C/EBPβ). Suppression of *Rik*-*201* and *Rik*-*203* restrained neural differentiation via a ceRNAs activity on miR-96 and miR-467a-3p, respectively, whose disinhibition restricts the expression of differentiation-related gene *Sox6* [81]. 

The lncRNA ***MEG3*** is upregulated by the cAMP/response element-binding protein (CREB) pathway to increase the expression of neuron-specific genes and act as a ceRNA, influencing the expression of miR-128-3p [82].

The lncRNA ***lncR492*** is a lineage-specific inhibitor of neuroectodermal differentiation in murine embryonic stem cells through interaction with the mRNA binding protein HuR and activation of the WNT signaling pathway [83].

lncRNAs can also directly regulate the splicing process such as the lncRNA ***Pnky***, expressed in the cortex, divergently transcribed from the proneural transcription factor POU3F2. In NSCs, *Pnky* and the RNA-binding protein PTBP1 regulate the expression and alternative splicing of a core set of transcripts that relate to neurogenesis [84,85]. The loss of *Pnky* does not affect the expression of POU3F2, but increases the neuronal lineage commitment of the NSCs [85,86], suggesting a suppressive function. Accordingly, the expression of *Pnky* decreases when ventricular and subventricular NSCs differentiate into neuronal cells.

The lncRNA ***MIAT*** has been shown to control the differentiation of neural progenitors and the survival of newborn neurons. In vivo experiments reported the pleiotropic effects on brain development and the altered splicing of Wnt7b after *MIAT* overexpression or RNAi silencing [87].

### 2.3. lncRNAs Harboring ORFs for Microproteins

Recent findings indicate that some lncRNAs contain small open reading frames (ORFs) that are in fact translated into small peptides [88,89,90]. These small ORFs have long been overlooked, mainly due to an arbitrary cutoff of 300 nucleotides set by ORF prediction algorithms. Nowadays, a wide variety of proteins smaller than 100 amino acids, called microproteins or SEPs (sORF-encoded peptides), have been discovered and characterized [91,92]. 

Microproteins have been shown to play key functions in a wide variety of cellular processes, from the mRNA turnover [93] to the DNA repair [94] and mitochondrial function [95]. 

The ***TUNAR*** lncRNA, also known as Megamind in zebrafish [96], is expressed during the early embryonic development [97], whereas in adults, it is mainly expressed in neurons and young oligodendrocytes [98]. *TUNAR* has been shown to maintain pluripotency and promote neural lineage commitment [99]. Specifically, it forms a complex with three RNA binding proteins, and such a complex binds to the pluripotency gene promoters *Nanog* and *Sox2*, and to the promoter of the neural differentiation factor *Fgf4* [99]. *TUNAR* has recently been shown to encode a 48 amino acid microprotein, named pTUNAR, detected in the nervous system, whose deficiency promotes neural differentiation of murine embryonic stem cells, whereas its overexpression impairs neurite outgrowth [95]. These findings underscore the importance of the microproteome as a source of previously undescribed regulators of neural differentiation.

Given the large number (tens of thousands) of predicted sORF peptides, several other microproteins are expected to play key roles in neurodevelopment, and thus in NDD pathoetiology.

## 3. lncRNAs in ID and Related NDD

Most lncRNAs in the human genome remain of unclear function but are expressed in the brain [100], and many of them are primate-specific [31]. Thus, we may expect that much of the lncRNA-mediated genetic information in humans is linked to brain function and development [101]. In the developing brain, lncRNAs are expressed at lower levels compared to protein-coding genes, and show high spatiotemporal specificity [32,87,102]. Structural alterations including amplifications, deletions, and translocations affecting the integrity of the lncRNAs have been reported to contribute to diseases such as cancer, AIDS, inflammatory bowel disease, and diabetes [103,104]. Evidence is also emerging of an association of lncRNA dysfunction with neuropsychiatric and neurodegenerative conditions [105,106].

Here, we focus on ID and classify ID-relevant lncRNAs based on their known, or proposed, functions (Figure 1; Table 1) [55,68,107,108,109].

### 3.1. Chromatin Remodeling and Transcriptional Regulation

lncRNAs regulate transcription through different mechanisms, among which there is the modulation of gene expression via the recruitment of chromatin-remodeling complexes and histone-modifying enzymes [13]. Several syndromic forms of ID were recently related to the altered expression of lncRNAs.

Fragile-X syndrome (FXS) (OMIM #300624) is the most common single-gene cause of ID, with about 1 in 4000 affected males and 1 in 7000 affected females [110,111]. The cause of the disease is an expanded trinucleotide repeat in the 5′ UTR region of the X-linked fragile X mental retardation 1 gene (*FMR1*) beyond the pathological threshold of 200 CGG/GCC repeats [112]. This leads to *FMR1* gene silencing by the methylation of the repeat that extends to the promoter, resulting in decreased FMRP levels in the brain [113]. Among the lncRNAs associated with FXS, ***FMR4*** is transcribed from the *FMR1* locus, functioning as a trans-acting chromatin-associated transcript for genes related to neural development and cell proliferation. *FMR4* overexpression changes the histone methylation status at promoters, genome-wide, by altering the H4K4me3 activation and H3K27Me3 repression marks. These changes result in the altered expression of several neurodevelopmental genes [114]. 

*FMR1* promoter also produces the antisense lncRNA transcript ***FMR1-AS1*** (OMIM #300805), highly expressed in the brain and kidney. *FMR1-AS1* is upregulated in patients with premutation alleles (55–200 repeats) and is not expressed in full mutation individuals. A contribution of *FMR1-AS1* to explain the variable phenotypes associated with FXS has been proposed [115]. 

*MECP2* mutations are causative of Rett syndrome (OMIM #312750) and specific forms of X-linked ID (OMIM #300055 and #300260). MECP2 binds the methylated CpGs and activates and represses transcription [116]. Recent studies have shown that mutations in *MECP2* can alter its ability to enhance the separation of heterochromatin and euchromatin through its condensate partitioning properties [117]. MECP2 activity is modulated by lncRNAs: in interneurons the lncRNA *Evf2* associates with MECP2 at the regulatory elements that control *Dlx5*, *Dlx6*, and *Gad1* expression, suggesting a possible role in the development of GABAergic neurons [69]. In *Mecp2*-null mice, the lncRNAs ***AK081227*** and ***AK087060*** were found to be upregulated and the increased levels of *AK081227* subsequently caused downregulation of its host coding protein gene, the GABA-receptor subunit RHO2 (GABRR2), adding further complexity to the mechanistic role of *MECP2* in Rett syndrome [118].

Among the lncRNAs involved in transcription regulation, ***DALI*** (DNMT1-Associated Long Intergenic) is an intergenic lncRNA transcribed downstream of the *Pou3f3* gene, able to control its expression. They share the same transcriptional targets, thus regulating a set of genes involved in neural differentiation. Chalei et al. showed that *DALI* can act in trans to control the transcription of distal genomic regions; in particular, it associates with active chromatin regions. Moreover, it associates directly with DNA methyltransferase DNMT1 (associated with autosomal dominant neurological disorders; OMIM #604121 and #614116), the BRG1 component of SWI/SNF complex, and the transcription factors P66beta and SIN3A, thus supporting its role as a regulator of chromatin-modifying proteins [119].

A lncRNA of interest in Down syndrome (DS) (OMIM #190685), is ***NRON***, which modulates the transport of Nuclear Factor of Activated T cells (NFAT) family proteins from the cytoplasm to the nucleus [120]. In a mouse model, reduced activity of NFAT caused DS-like features including increased social interaction and locomotor activity, decreased muscular strength, and anxiety-related behavior [121]. It is therefore tempting to speculate that impairments in *NRON* could be involved in DS etiology. 

Other lncRNAs were found to play crucial roles in the pathogenic mechanisms behind other forms of syndromic ID such as in Mowat–Wilson syndrome (OMIM #235730), a disorder associated with single-nucleotide variants in the *ZEB2* gene. *ZEB2* expression is indeed controlled in several manners including tissue- and time-specific regulation by an antisense lncRNA named ***ZEB2-NAT***. The mechanism involves the retention of the first intron of *ZEB2* pre-mRNA [122,123]. Further studies are needed to clarify this regulation.

Microphthalmia Syndromic 3 (MCOPS3; OMIM #206900) is a syndromic disorder that includes learning difficulties and psychomotor delay. The lncRNA ***RMST*** is regulated by the transcriptional repressor REST and increases during neuronal differentiation. It physically associates with the transcription factor SOX2 and together, they regulate the downstream pathways involved in neurogenesis [63]. 

### 3.2. Signaling and Transduction 

The WNT, NOTCH, and MAPK signaling pathways are critically required during brain development. Their alteration has been implicated in neuropsychiatric disorders including ASD and ID [124,125]. WNT signaling regulates cell proliferation, cell fate determination and differentiation during both embryo development and tissue homeostasis in adult individuals [126]. Mutations affecting the canonical WNT pathway have been linked to neuron loss and deficits [127]. NOTCH signaling is a key pathway involved in regulating the balance between stem cell maintenance and proper neuronal differentiation [128]. 

The lncRNA ***GAS5***, previously associated with cancer [129], Klinefelter syndrome [130], and autoimmune diseases [131], has been linked to DS [132]. *GAS5* plays an important role in brain development, controlling the apoptosis, migration, and proliferation of neural cells through the regulation of *GSTM3* and thus the NOTCH pathway [133]. In a limited cohort of 23 DS patients, a *GAS5* downregulation was reported, suggesting a possible dysregulation of its downstream pathways [132]. Interestingly, other studies have reported that the dysregulation in lncRNAs expression in induced pluripotent stem cells (iPSCs) derived from DS patients was even more significant than that of mRNAs [132,134]. In another study on a small cohort of patients with Klinefelter syndrome, the authors showed a 5-fold increase in *GAS5* [130], suggesting that an alteration in its levels can be involved in different disorders. 

Another lncRNA regulating the NOTCH pathway is ***LncND***, which has been found in a microdeletion at 2p25.3, causative of ID. *LncND* is highly expressed in neural progenitors and has been shown to act as a miRNA sponge sequestering miR-143-3p, thereby controlling NOTCH-1 and NOTCH-2 expression. *LncND* knockdown experiments resulted in premature differentiation of neural progenitors, whereas its overexpression in mice—although this lncRNA is selectively expressed in primates—confirmed the involvement of *LncND* in maintaining the neural progenitors’ pool during cerebral cortex development [135].

*Maternally Expressed Gene 3* (***MEG3***) is a lncRNA found highly expressed in the forebrain neurons of rodents [136]. In a rat model of cerebral ischemia-reperfusion injury, *MEG3* knockdown was able to ameliorate neurological impairment, reduce neural apoptosis and necrosis, and enhance neurogenesis through activation of the WNT pathway [137]. Thus, the correlation of this lncRNA with neuronal impairment is worthy of further investigation to assess its possibility as a therapeutic target.

*Nuclear Enriched Abundant Transcript 1* (***NEAT1***) has been shown to increase the protein level of the WNT signaling factors, affecting reperfusion and neuroinflammation injury [138]. Moreover, the miR-124-Neat1-WNT/β-catenin signaling axis was found to induce neuronal differentiation, inhibit apoptosis, and promote the migration of murine spinal cord progenitor cells. The overexpression of miR-124 enhanced the expression of *NEAT1*, resulting in the amelioration of spinal cord injury in the mouse model [139]. *NEAT1* has also been related to the MAPK/ERK pathway [140]. In the same work, the expression of *LincPint* was found to be regulated by the ERK pathway. *LincPint* interacts with Polycomb repressive complex 2 and promotes cell survival [141].

***MALAT1*** expression profiling data from in vitro differentiation of Neuro-2a neuroblastoma cells showed that *MALAT1* was significantly upregulated among other lncRNAs during differentiation [74]. Its knockdown resulted in the enhanced repression of neurite outgrowth and a reduction in cell survival; among the affected pathways, MAPK/ERK emerged as the most inhibited one, thus Chen et al. suggested that *MALAT1* promotes neuronal differentiation and survival through the activation of the MAPK/ERK signaling pathway. In contrast to these results, a more recent work from Shi et al. showed that *MALAT1* inhibits the MAPK/ERK pathway, leading to increased apoptosis in the cortical tissues in models of cerebral infarction [142].

### 3.3. Synaptic Functions 

The lncRNA ***BC200***, which shares tissue and cellular localization with the Brain Cytoplasmic 1 (BC1) gene [143,144], regulates synaptic plasticity and excitability through the repression of synapse local translation, interacting with FMRP and recruiting it to target mRNAs [145,146]. *BC1/BC200* also binds the poly(A)-binding protein (PABP), preventing the translation initiation of other polyadenylated mRNAs in neurons [147]. Moreover, the *BC200* lncRNA interacts with the RNA-binding protein SYNCRIP via an internal A-rich region to regulate dendritic mRNA transport and synaptic plasticity in human neurons [148].

***neuroLNC*** is a conserved lncRNA whose expression is restricted to the brain and goes from the developmental period to adulthood. *neuroLNC* interacts with coding mRNAs related to glutamate receptor signaling, the regulation of membrane potential, and synapse organization [149]. Moreover, it also affects synaptic release by binding to TAR DNA binding protein-43 (TDP-43), which is able to bind the synaptic vesicles/endocytic proteins [150]. It has been suggested to be a central player in the pathogenesis of neurodegenerative disorders such as amyotrophic lateral sclerosis and frontotemporal dementia [151].

The nucleolus-specific lncRNA ***LoNA*** has been shown to regulate specific subunits of the N-methyl-D-aspartate receptors (NMDARs), which are glutamate-gated ion channels regulating neuronal communication and synaptic function. *LoNA* inhibits rRNA production and ribosome biosynthesis, and consequently protein synthesis at the resting state. It acts by binding to nucleolin, which has been shown to alter the epigenetic state of rDNA and modulate rRNA biosynthesis. *LoNa* knockdown leads to the alleviation of rRNA synthesis suppression, increases the AMPA/NMDA receptor level, enhances neuronal plasticity, and improves the LTP and long-term memory. Interestingly, the NMDAR subunits NR1, NR2A, and NR2B, all associated with NDD (OMIM #617820, #614254, #245570, #616139, #613970), were upregulated in this model [67].

***Synage*** is a lncRNA highly expressed in the cerebellum where it regulates synaptic stability via at least two mechanisms: acting as a sponge for miR-325-3p, to regulate the expression of the cerebellar synapse organizer *Cbln1*, and as a scaffold for organizing the assembly of the LRP1-HSP90AA1-PSD-95 complex in parallel fibers-Purkinje cell synapses [152].

***GM12371*** has a role in maintaining the excitatory postsynaptic transmission in hippocampal neurons and is essential for dendritic arborization and spine complexity; it is a trans-acting regulator of genes involved in neural growth and development such as *Soxo10* and *PRKCq* [153]. 

### 3.4. Cytoskeleton Dynamics and RHO-GTPases 

*SYNGAP1* is associated with Mental Retardation Autosomal Dominant 5 (OMIM #612621), a disorder that presents mild to severe ID, developmental regression, and epilepsy. The antisense RNA corresponding to *SYNGAP1* locus, ***SYNGAP1-AS***, is upregulated in the postmortem brains of ASD patients, particularly at the level of the prefrontal cortex and superior temporal gyrus [154]. SYNGAP1 inhibits the RAS GTPase cascade in neurons and was shown to be important for the regulation of the AMPA receptor trafficking and neuritogenesis [155,156]. It is therefore likely that *SYNGAP1-AS* is involved in the same pathways and biological processes.

The lncRNA ***MSNP1AS*** was found to be overexpressed in the postmortem cerebral cortex of ASD patients characterized by the 5p14.1 marker [157]. Recently, it was shown that it can suppress the level of the cytoskeletal protein moesin by binding to its sense RNA. Overexpression of *MSNP1AS* causes the activation of the RHOA pathway and the inhibition of the RAC1 and PI3K/AKT pathways [158].

### 3.5. Other Functions 

Functions of most lncRNAs are hardly predictable or classifiable into defined biological processes. The 5q14 locus has been suggested to be involved in NDD. The region encompasses several annotated genes, among which is the lncRNA ***lnc-NR2F1*,** a highly conserved sequence across species [159]. *lnc-NR2F1* is disrupted by chromosomal translocation t(5:12) in a family with NDD symptoms [68]. Furthermore, loss- and gain-of-function studies in mouse ES cells demonstrated that *lnc-NR2F1*, but not the nearby protein-coding gene NR2F1, is needed for neuronal cell maturation and acts by regulating the transcription of a set of neuronal genes [68]. *lnc-NR2F1* participates in neuronal migration pathways, and it has been shown to regulate genes implicated in NDD [68]. Notably, the proband’s father, who suffered from dyslexia and stuttering, carried the same translocation, suggesting a possible Mendelian inheritance, with clinical variability [68]. To date, only one family has been identified with *lnc-NR2F1* disruption; therefore, the identification of additional families will be required to confirm that *lnc-NR2F1* is responsible for an NDD. 

Chromosome 2p16.1-p15 deletion syndrome (OMIM #612513) is an NDD characterized by ID, delayed psychomotor development, and variable dysmorphic features. A 0.45 Mb deletion encompassing three genes (*BCL11A*, *PAPOLG*, and *REL*) and the lncRNA ***FLJ16341*** was found in a patient [160]. This region overlaps with others reported for patients with similar phenotypes [161]. The function of this lncRNA is currently unknown.

Additionally, the disruption of ***LINC00299*** has been linked to neurodevelopmental delay. The disruption of 2p25.1 has been identified in cases of NDD, while no structural variants have been observed in a control cohort of about 14,000 cases. In a patient with a history of speech delay, mild ID, bipolar disorder, and epilepsy, a minimal deleted region of approximately 60 kb that did not alter neighboring genes has been reported. In another patient affected by a disorder resembling Angelman syndrome, disruption of *LINC00299* has been found [162]. The *LINC00299* had increased levels in the patient’s lymphoblastoid cell line, and no disruption of this lncRNA has been found in control subjects [162]. It is therefore possible to suspect a direct involvement of this lncRNA in NDD.

Finally, in a study on 25 patients with ASD, performed on peripheral leukocytes, the authors reported almost 4000 differentially expressed lncRNAs. Among them, nine upregulated and four downregulated lncRNAs were involved in metabolic pathways, synaptic vesicles and synaptic plasticity [108]. In another work on the post-mortem brains of autistic patients, 71 natural antisense transcripts (NATs) were identified as associated with ASD including *FOXG1-AS* and *SYNGAP1-AS.* Most of the identified NATs were involved in mechanisms of chromatin modifications or in transcriptional regulation [154].

## 4. The Search for Relevant lncRNAs Using Integrated Approaches

Overall, a causative association between the over- or under-expression of specific lncRNAs and the pathologies indicated above is still widely speculative. On the other hand, we should consider that a large fraction of patients with sporadic ID still lack a genetic or molecular explanation, and therefore other disease mechanisms should still be sought. The few examples illustrated above are likely to represent the tip of the iceberg, and additional studies may prove that lncRNA dysregulation is a prominent mechanism underlying human ID and related disorders. 

To forge ahead in the search for lncRNAs relevant for the comprehension of the ID endophenotype, integrative approaches that combine the analysis of “omic” wet data with existing databases and bioinformatic predictions will become very useful. Although, to a large extent, the approaches that we will illustrate are still only predictive, they set the frame for experimental work. All of the approaches listed below have to be considered complementary to one another and not mutually exclusive in the search of ID- and NDD-risk lncRNAs candidates. 

### 4.1. In Vitro Functional Screening Based on Differentiation Programs and In Vivo Transcriptome Analyses

During organ development, lncRNAs play several roles in establishing and maintaining cell-type-specific gene expression patterns. Considering the human brain, this is by far the most complex and diversified organ in terms of the cellular types, neuronal and glial. This complexity is reflected in its great number of unique cell-type-specific lncRNAs and was linked to the evolution of the human brain itself, in terms of its morphology and functions [163,164,165].

The expression of a large number of lncRNAs has also emerged in vitro before and during the differentiation processes by, respectively, promoting the pluripotency and neuronal differentiation and maturation [166]. The microarray analysis showed a coherent correlation between the expression of several lncRNAs and neural specification protein-coding genes, whereas in situ hybridization revealed their association in a transcriptional complex with differentiation-specific nuclear subdomains (e.g., *Gomafu* and *NEAT1*) [32]. 

Stem cell differentiation and direct cell reprogramming are attractive systems for studying the function of lncRNAs, as the analysis of the expression profiles at different time points can prove their involvement in cell fate establishment [167]. An example of this approach is provided by the study that led to the identification of *lnc-NR2F1* as a risk non-coding genomic locus for ID and ASD [68]. This study incorporates high-throughput cell fate reprogramming, human genetics, and the functional analysis of lncRNAs. Indeed, lncRNA candidates were first annotated for being differentially expressed in the direct lineage conversion from the mouse embryonic fibroblast to neuronal cells; these were further selected by intersecting the human genomic coordinates of their loci with a co-morbidity map derived from a large cohort of ID/ASD patients (n = 29,085) [168,169] and validating the final candidate loci through a custom tiling array. The *lnc-NR2F1* was molecularly and functionally characterized to investigate its contribution to neurodevelopment. To date, only one family with lnc-NR2F1 disruption has been identified, therefore, the identification of additional families will be required to confirm *lnc-NR2F2* as being responsible for the phenotypes.

Analogously, the RNA sequencing (RNA-Seq) of in vitro differentiated neurons derived from human iPSCs revealed how several lncRNAs that increased during the transition also mapped near the single nucleotide polymorphisms (SNPs) associated with schizophrenia in genome-wide association studies (GWAS), thus suggesting their possible misregulation in a subgroup of patients [170]. Regarding the in vivo approaches, they need to be complemented with extensive meta-analysis. An optimal initial selection of possible candidates can be applied by using transcriptome-derived data by comparing different cells at different neural stages (i.e., proliferating NSCs, progenitors, and newborn neurons) [87] or by looking at differentially expressed lncRNAs in samples of the human brain at crucial ages for brain development (i.e., between 3 weeks post-conception and 8 years of age), also by using appropriate datasets such as the BrainSpan developmental transcriptome dataset [171].

Microarray and RNA-Seq of tissue samples derived from both ASD and major depressive disorder patients have also been extensively used to identify differentially expressed lncRNAs, that, in the absence of gene mutations, rearrangements, or losses, seem to be relevant biomarkers or else causative factors of such diseases [107,108,154,172].

### 4.2. Genome-Wide Association Studies

Coding-gene mutations explain only a small portion of Mendelian inherited disorders, whereas many others still need to be solved in terms of their genetic etiology. Such complex diseases (i.e., CNS, cardiovascular, immune disorders, diabetes, and cancer) present complex phenotypes and often an unpredictable drug treatment outcome [173]. Non-coding RNAs are now largely considered as genetic factors contributing to disease risk, especially with regard to rare diseases [174]. 

The purpose of GWAS is to associate pathological/phenotypical traits to specific genetic variants observed in different individuals by investigating the entire genome, in contrast with methods that specifically focus on pre-specified, often exonic, regions. The “common disease-common variant” is the hypothesis on which GWAS is based, predicting that common-disease variants are at least present in 5% of the entire population [173,175]. However, the most common variants are not usually sufficient to fully explain the heritability of common traits or the risk increment associated with complex diseases, as shown by the small effect sizes of the individual variants associated with the human height, whose heritability was previously estimated to be around 80% [176].

GWAS frequently focus on SNPs; these are likely to find a high density of SNPs in the human genome, most of which do not lie within protein-coding sequences, highlighting a possible role for the non-coding genome in disease complex traits etiology [173]. Concerning lncRNAs, it is possible to envision the effect that SNPs can have on their functionality, both directly and indirectly. A direct effect refers to variants found in the lncRNA sequence itself that can alter sequences that are crucial for the complementarity with the binding partners (e.g., other nucleic acids), constitutive/alternative splicing, or folding (e.g., hairpin loop formation), and thus stability [103]. An example of a study that has deepened conformation-altering RNA polymorphisms was provided by Mirza et al., as it explored the attributes of candidate lncRNAs for inflammatory bowel disease and type 1 diabetes, predicting the structural effects of mapped SNPs within them [177]. Otherwise, a possible polymorphism indirect effect on the lncRNAs functionality could be exerted by disease-associated SNPs found in lncRNA regulatory promoter elements [178,179], which may perturb the transcription factor recognition sequences, altering their binding and thus affecting lncRNA expression. 

SNPs at the lncRNA loci have been associated with both phenotypical and pathological conditions, ranging from bodyweight [180,181,182] and brain gray matter volume in the normal population [183] to several types of cancer [184,185,186,187,188]. However, SNP associations account for only a fraction of the genetic component of most diseases.

GWAS also take into account the role of rare copy number variants, either individually or in combination, in an increased risk for neurodevelopmental diseases [189,190,191]. Meng et al. examined lncRNAs within 10 schizophrenia risk-associated CNV deletion regions and examined their potential contribution to the disease, finally identifying a promising candidate—the lncRNAs *DGCR5*—as a potential regulator of certain schizophrenia-related genes [192].

Non-coding variants are already included in polygenic risk scores for multiple complex diseases such as breast cancer [193], but most predictions have been derived from analysis of European ancestry, thus highlighting the limit imposed by ethnic groups on GWAS.

### 4.3. Gene Co-Expression Network Analysis

The co-expression relationships derived from the microarray or RNA-Seq data represent an important source of information, potentially relevant for functional annotation and disease gene prediction. Indeed, it has been shown extensively that functionally interacting genes tend to show very similar expression profiles as a result of common regulatory mechanisms [194,195], thus gene co-expression networks allow for the analysis of expression datasets, considering not only one gene at a time in a reductionist approach, but tackling the complexity of the considered system as a whole.

The most common approaches consist of defining modules: groups of highly correlated genes, based on expression profiles in a tissue or condition of interest, which likely share genetic regulation and/or biological function. By clustering genes into co-expression modules, the biological function of a lncRNA may be inferred from the Gene Ontology (GO) enrichment of the known genes in the module and the degree to which the lncRNA correlates within the module.

A large number of lncRNAs are differentially expressed during neural development and appear to be strongly linked to high-risk ID and ASD genes in co-expression networks [125,196], but very few have been molecularly and functionally characterized until now.

For the sake of the prioritization of lncRNAs, a first approach was adopted by Gudenas et al. [125]. The authors curated a list of lncRNAs likely to participate in ID based on three main criteria: (1) Their putative involvement in neuronal development, based on their differential expression in developing human neocortex samples; (2) the correlation with known ID-risk genes as measured by weighted co-expression networks analysis approach [197] using gene expression profiles measured in the same samples; and (3) their location in the CNV regions of ID patients. More than 500 lncRNAs were found to satisfy these criteria and are therefore likely to play a role in ID pathogenesis or severity, which is a realistic number, but still too high to be approached experimentally.

Gene expression networks constructed on healthy adult samples can be useful to bioinformatically predict the roles of lncRNAs in diseases. An example of this approach is the FuncPred tool [198], which analyses the networks constructed starting from thousands of RNA-Seq samples provided by the Genotype-Tissue Expression (GTEx) project [199] and can be used to understand the possible role of a lncRNA in a physiological or pathological process. FuncPred provides putative associations between the lncRNAs and keywords of different ontologies including GO and Disease Ontology. The GO term association provided by the FuncPred algorithm can also be used to obtain evidence of the possible involvement of some lncRNAs in cellular core regulatory processes whose misregulations are linked to the ID.

### 4.4. Other Selections Based on In Silico Predicted Functions

Further selection and prioritization may be obtained by the study of known, or putative, regulatory targets of a lncRNA [200], thus deepening its molecular functions. Simple models include the cis-action of antisense-lncRNAs as repressors of protein-coding genes of the same genomic locus or encoded in genomic proximity. More complex models include the trans-action of the lncRNA for which the prediction of putative targets may be more complex. One of the possible mechanisms theorized is the Watson–Crick base pairing between the lncRNA and the target messenger RNA (mRNA); specifically, this class of lncRNA is named as trans-natural antisense transcripts and can be detected when looking for RNA sequence complementarity [201].

A more studied trans-acting mechanism, named the miRNA sponge, involves the interaction of lncRNAs, miRNA, and mRNA. In this case, the lncRNA sequence presents binding sites for specific miRNAs, thus competing for their binding with mRNAs harboring the same binding sites, resulting in a positive regulation of mRNAs through the sequestration of the repressing miRNAs [202]. An updated resource reporting known miRNA–lncRNA interactions is DIANA-LncBase v3 [203]; Zang et al. provided an example of a modular approach to identify a lncRNA-related miRNA sponge in breast cancer [204].

Another emerging type of gene regulation involving lncRNAs requires the formation of RNA:DNA:DNA triple helices through Hogsteen base pairing in cis or in trans [11,205]. Algorithms have been developed to identify putative genomic regions that can form triple helices with a given lncRNA [206,207].

Moreover, ENCODE [208] and other initiatives [209,210] provide extensive information on the chromatin states in various tissues and conditions including different neuron types and progenitors or brain regions. The enrichment of regulatory regions predicted to be bound by a lncRNA in an open chromatin configuration, related to a specific condition, can suggest the involvement of the lncRNA in regulatory processes relevant to that condition itself. By linking bound regulatory regions to target genes, further information can be putatively inferred [211]. Unfortunately, the experimental data using ChIRP [212], CHART [213], or other techniques developed to identify the DNA binding sites of the RNA molecules only exist for a few lncRNAs [214].

## 5. lncRNA in Human Diseases: Tools, Platforms, and Applications in Diagnostic

Considering the wealth of in vitro data showing relevant functions of lncRNAs for the commitment, proliferation, and differentiation of young neurons, and considering the growing body of evidence of the involvement of lncRNAs in human diseases, the importance of lncRNAs as causative or contributors to NDD seems established. However, the involvement of altered lncRNA functions in human ID and related NDD remains poorly explored. One of the main reasons is related to the tools used in diagnostic settings. 

Molecular diagnoses for ID patients are routinely performed by array-CGH/SNP-array analyses, searching for large genetic deletions and duplications [215]; even if these methods are, in principle, able to identify the involvement of lncRNAs, the analysis is limited to coding genes because of the lack of knowledge of pathogenicity for lncRNAs. In a second step, gene panels or exome sequencing are often performed [216]. These strategies are again focused on the protein-coding regions of the genome, and non-coding transcribed sequences are discarded. In about 13% of NDD and ID patients, array-CGH detects a pathogenic CNV, possibly having a causative link with the disease phenotype [215]. Exome sequencing increases the rate of a positive diagnosis by 4–17% for non-syndromic ID, while it significantly increases up to 62% for syndromic ID [216]. Currently, a similar diagnostic rate has also been reported for genome sequencing [217], which should include lncRNAs. However, we must consider the lack of appropriate bioinformatics tools for the analysis of lncRNAs, which may highlight potentially pathogenic variants and infer altered lncRNA function based on altered lncRNA sequence. The possibility that the altered nucleotide sequence or the level of expression of lncRNAs might represent a relevant cause of ID and related NDD calls for a new layer of complexity in the molecular architecture of ID, and, at the same time, opens new opportunities for their use as diagnostic markers.

Beyond the difficulties in sequence interpretation, the validation of a lncRNA variant is not straightforward, since there are no direct methods to estimate the changes in the expression levels in the patients’ brain cells. Thus, the translation of new findings in the role of lncRNAs in human neuron biology toward diagnostic innovations is lagging behind. 

Whole-genome sequencing is certainly the way to go in these cases in which whole-exome sequencing has previously failed to identify the most-likely causative allelic variant(s). The results of the genome sequencing of patients should then be compared with the data derived from a large cohort of healthy individuals. To further investigate the significance of specific variants in patients, we must rely on the intelligent use of data banks (e.g., MSSNG portal for ASD, https://research.mss.ng/). The lncRNAs’ polymorphism and disease association are currently provided by a small number of specialized databases (a summary is provided in Table 2).

Another possibility could be to determine the expression level of key lncRNAs. Currently, lncRNAs have been proposed as novel targets for the diagnosis and treatment of diabetes [218] and cancer [219,220]. Particularly in cancer patients, lncRNAs can be detected in body fluids including whole blood, plasma, serum, urine, saliva, and gastric juice and display a dynamic alteration in the presence of the diseases [221,222,223]. The collection of these fluids is non-invasive and inexpensive, thus they are being considered in diagnosis [224]. Regarding NDD, the use of lncRNAs as biomarkers for diagnosis or treatment is far from coming to fruition, although six lncRNAs have been suggested as potential biomarkers for diagnosis and therapy in major depressive disorder [225]. 

As lncRNAs expression cannot be detected from the patients’ neurons, the only two alternatives adoptable are: (1) fibroblasts (or similar easily obtainable cells) from the selected patients to be reprogrammed to iPSC, differentiated in neural cell types, and then used for RNA-Seq expression studies, or (2) the expression could be determined in non-neuronal cells such as blood mononucleated cells to establish biomarkers to be used as a reliable signature in large screening studies. Any new findings of altered sequence or expression level associated with the ID in humans need experimental validation to confirm a causative role. For this reason, functional genetics in animal models and human iPSC are still needed and represent the most time- and money-consuming step.

The importance of searching for possible non-coding biomarkers and diagnostic elements in ID and related NDD is supported by these expected outcomes: (i) the possibility of obtaining information regarding the genotype and/or the response to treatment from a non-invasive fluid; (ii) the opportunity to solve the portion of cases who did not obtain a diagnosis from the aforementioned methods (array-CGH, ES, GS); and (iii) the perspective of defining, or discovering, new molecular pathways of regulation for disease-related genes. In the near future, it is fair to imagine an integration of array-CGH, ES, GS, and RNA-Seq to identify the coding and non-coding pathogenic variants, increasing the diagnostic rate and reducing the time needed for a diagnosis [226]. 

**Table 2 ijms-23-06118-t002:** The databases reporting lncRNAs polymorphism and association with disease.

DATABASE	WEBSITE	REFERENCE
**LINCSNP 3.0**	http://bio-bigdata.hrbmu.edu.cn/lincsnp/index.jsp	[227]
**LNCVAR**	http://159.226.118.31/LncVar/	[228,229]
**LNCRNADISEASE V2.0**	http://www.rnanut.net/lncrnadisease/	[230,231]
**RNACENTRAL**	https://rnacentral.org/	[232]
**EVLNCRNAS V2.0**	http://biophy.dzu.edu.cn/EVLncRNAs/	[233]
**NONCODE**	http://noncode.org/	[234]
**LNCBOOK**	https://ngdc.cncb.ac.cn/lncbook/index	[235]

## 6. Concluding Remarks and Perspectives

For decades, we have concentrated on the coding part of the genome, leaving the large non-coding, but still transcribed, portion aside. Attempts to cluster the ID-risk coding genes into functional categories have led to the recognition of a limited number of neuronal hubs involved in the pathology endophenotype [51]. Still, a significant fraction of patients with sporadic, syndromic, or non-syndromic ID lacks a genetic and/or molecular explanation. 

Among the non-coding portion of the genome, lncRNAs are highly expressed in the human brain, both in normal and pathological conditions; however, their involvement in NDD, and specifically in ID, is still poorly documented due to the underestimation of lncRNA mutations or altered expression in human disorders. Times are rapidly changing, and considerable efforts and advancements are now expected to lead to the development of improved bioinformatic tools dedicated to the investigation of lncRNAs. 

The translation of new findings on the role of lncRNAs in the biology of human neurons toward diagnostic innovations is conceivable. Whole-genome sequencing could be conducted in those cases in which exome sequencing has previously failed to identify causative allelic variant(s) and the results should then be compared with the data derived from a large cohort of healthy individuals. Extensive data banks can be used to further investigate the significance of specific allelic variants in patients. Moreover, the expression levels of key lncRNAs should be determined in reliable models such as neurons derived from the reprogrammed fibroblasts or peripheral blood mononuclear cells. Alternatively, a specific set of lncRNAs could be identified and used as signatures in screening methods, to be applied on readily available samples such as liquid biopsies. 

In addition to the upcoming use of lncRNAs for diagnosis and prognosis, the targeting of lncRNAs for therapy is in progress concerning cancer, cardiovascular, neurological, and muscular diseases. In the future, RNA-based strategies are expected to offer new clinical opportunities in the symptomatic treatment of NDD and, in particular, ID, although the high variability of the molecular dysfunctions will be the main obstacle. The choice of validated and informative cell-based and in vivo models will be essential.

## Figures and Tables

**Figure 1 ijms-23-06118-f001:**
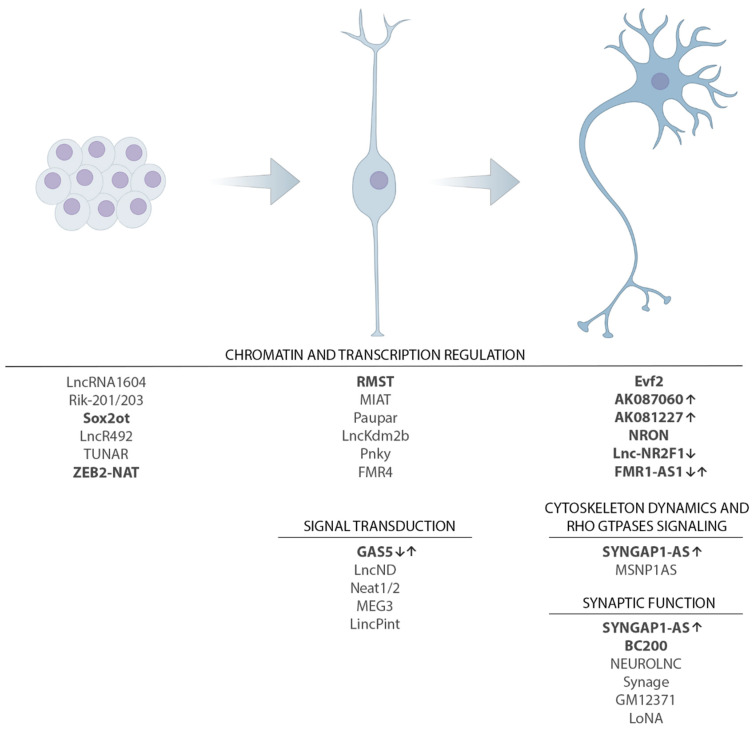
The lncRNAs involved in the sequential stages of developing neurons. The scheme illustrates three ideal key stages in the commitment/differentiation of developing neurons, from stem cells (**left**) to neural progenitor cell (**middle**) and to postmitotic differentiating neuron (**right**). At the bottom, a few representative lncRNAs are reported for each stage, with an indication of the intracellular function or process in which it has been involved. lncRNAs with a possible implication in NDD pathogenesis are reported in bold; the up and down arrows indicate, respectively, increased and reduced expression of lncRNAs in post-mortem biopsies or models of NDD.

**Table 1 ijms-23-06118-t001:** The lncRNAs implicated in the etiology of ID.

*lncRNAs Directly Involved in ID*				
lncRNA	Observed Anomaly	Associated Phenotype Observed		Mechanism/Evidence
** *lnc-NR2F1* **	[t(5:12)]	Developmental and speech delay		Disruption of *lnc-NR2F1*, which controls neuronal migration and other NDD-genes
** *LINC00299* **	2p25.1 disruption	Speech delay, ID, bipolar disorder, epilepsy, Angelman-like syndrome		*LINC00299* increased levels in patients
** *lncRNAs regulating genes involved in ID* **				
**lncRNA**	**Regulated gene**	**Possible association with disease**	**OMIM disease**	**Mechanism/Evidence**
** *BC200* **	*FMR1*	Fragile-X syndrome	#300624	Repression of local translation, interacting with FMRP
** *GAS5* **	*GSTM3*	Down syndrome	#190685	Downregulation of GAS5 in DS patients Upregulation in Klinefelter syndrome patients
** *NRON* **	*NFAT*	Down syndrome	#190685	Reduced NFAT causes DS-like phenotype; *NRON* modulates NFAT activity.
** *EVF2* **	*MECP2*	Rett syndrome X-linked intellectual developmental disorder	#312750 #300055 #300260	*EVF2* associates with MECP2 at regulatory elements in interneurons, controlling *DLX5*, *DLX6* and *GAD1* expression
** *AK081227* ** ** *AK087060* **	*MECP2*	Rett syndrome X-linked intellectual developmental disorder	#312750 #300055 #300260	Upregulated levels in *Mecp2*-null mice; *AK081227* downregulates *Gabrr2*
** *ZEB2-NAT* **	*ZEB2*	Mowat–Wilson syndrome	#235730	*ZEB2-NAT* controls *ZEB2* by retaining the first intron of *ZEB2* pre-mRNA
** *RMST* **	*SOX2*	Microphthalmia and optic nerve hypoplasia and abnormalities of the central nervous system	#206900	*RMST* physically associates with SOX2 regulating neurogenesis pathways
** *Sox2ot* **	*SOX2*	Microphthalmia and optic nerve hypoplasia and abnormalities of the central nervous system	#206900	*Sox2ot* represses *SOX2* RNA levels
** *SYNGAP1-AS* **	*SYNGAP1*	Mental Retardation Autosomal Dominant 5	#612621	*SYNGAP1-AS* is upregulated in post-mortem brains

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
