# Peer review of "The Emerging Roles of Long Non-Coding RNAs in Intellectual Disability and Related Neurodevelopmental Disorders"

_ijms, 2022, doi:10.3390/ijms23116118_

Round 1
Reviewer 1 Report
The review manuscript by Liaci et al is a fairly extensive compendium of known and suspected long non-coding RNAs in mental and neurodevelopmental disorders, and also discusses potential avenues for future explorations of this important emerging field with many unknowns. The review is well written, if a little on a long side, and appears to hit on all the important points. I may have learned a lot from reading it. I have no major concerns and my minor ones are mostly about clarity, as outlined below,
Line 2 (title). Should this be roles (in plural) rather than one role? The same apples to several places in the text (lines 574, 691for example) as well, starting with line 16 in Abstract.
Line 13 – remove “relevant”
Line 23 – what is “clustering candidates”?
Line 35 – should be tissues (plural)
Line 37 – delete “not contemplating their functional relevance”
Line 39 – delete “all”
Line 78 – delete “possibly”
Line 93 – exome sequencing (no hyphen)
Lines 98-101 – the sentence appears too long and hard to understand
Lines 108, 112 – check paragraph indents
Line 241 – delete the comma after “translocations”
Line 278 – “adding complexity on the mechanisms” sounds awkward
Line 284 – maybe split the sentence into two
Line 297 – “single nucleotide” should be single-nucleotide (hyphenated)
Line 301 – Check the name of the disorder. If this is the long name, may want to run it in Title Case.
Line 368 – delete the comma before “that”
Line 407 – insert “is” between “which” and “the”
Line 418 – should be “an NDD”
Figure 1 – is it possible to make the cell images smaller and the informative part (RNA names) larger?
Line 444 – “cannot but exist” – check wording
Line 469 – “cell direct reprogramming” should be “direct cell reprogramming”
Line 470 - “profile” should be “profiles”
Line 488-489 – “they have also to be complemented” – double check wording. Also, I am not sure what is meant by extensive meta-analysis. In vitro data often have a goal to understand molecular mechanisms, which is not always possible to supplement with in vivo/patients data (if that’s what is meant by “meta-analysis”)
Line 489-490 – unclear what the sentence is meant to say. What is “integrity of their genomic loci”?
Line 502 “in their genetic etiology” should be “in terms of their genetic etiology” – double check?
Line 508 – “individuals investigating” should be “individuals by investigating” – double check?
Line 535 – “take also” should be “also take”
Line 558 “is” should be “are” (the – is; a – are”)
Line 562 – “cured” should be “curated”
Line 602 – “state” should be “states”?
Line 613 – “implication” should be “involvement” and “the role of” should be “the importance of” – double check?
Line 617 – “is” should be “are”
Line 646 - what does “alternatively” apply to?
Line 653 – “coming” should be “coming to fruition” – double check?
Line 669 – “discover” should be “discovering”
Line 685 – “in human the brain” should be “in the human brain”
Line 696 – “level” should be “levels”
Author Response
We thank the reviewer for the helpful comments.
We have corrected all the errors that were indicated.
We have revised Figure 1 according to the suggestion. The changes also consider the comments from Reviewer 2 about the figure.
Reviewer 2 Report
Comments to authors:
The manuscript that was written by Dr. Liaci et al. is interesting to discuss the possible roles of non-coding RNAs (ncRNAs) in the regulation of brain development. Moreover, authors reviewed the ncRNA-mediated molecular mechanisms that could affect the intellectual disability (ID). However, it is obscure whether the expression of ncRNAs is corelated with the pathogenesis of the ID. Before publishing the article, I recommend authors extend discussions on the biological systems that affect up- and down-regulation of the ncRNAs. That may explain the dysregulation of the expression of ncRNAs could be originated from the aberrant profile of the transcription factors. Additionally, I recommend authors to discuss more profoundly on the causal relationships between the specific lncRNAs and diseases.
General comments
In this review article, authors described the essential information of the ncRNAs, discussing their roles in neuronal development and ID pathogenesis. Although it could provide insights in clinical studies in some extent, I would recommend authors more discuss on the mechanisms that regulate production of the ncRNAs that might affect neural development. I guess some transcription factors and epigenetic regulations, including DNA-methylation and histone modifications, are involved in the regulation of ncRNA expression. If so, differences in the level of the ncRNAs would be one of the results of the alteration in unknown key regulator(s). If not for that kind of discussions, it would not sound comprehensively. In addition, readers will naturally expect the discussion on molecular mechanisms to cause neural disorders that are mediated by specific ncRNAs. Authors might better pay more attention and focus on the studies that concerned with the comparison of the expression level of ncRNAs of patients and healthy doners. Or are there any reports that indicated mutations in some specific ncRNAs in families of ID patients? That may indicate the direct causative role of the ncRNAs on the pathogenesis of the ID.
Minor comments
Page 4, L172: “lncRNAa acting as post-transcriptional regulators” should be typed in italic but not in bold.
Page 7, L323 and 324: Please check the size and the style of the sentence.
Page 10, Figure 1: Several lncRNAs are classified by their functions. It should be indicated whether they were identified in HUMAN or in Mouse. Moreover, it can be shown by arrows whether up- or down-regulation of each lncRNA is associated with the morphology of a cell.
Page 10, Table 1: Authors summarized ncRNAs that could be associated with neural development and etiology of ID. It would be a great help for readers comprehension if it were shown separately according to nucleotide sequences, up/down regulation of expression of the ncRNAs.
Author Response
We thank the reviewer for the helful comments and suggestions.
Concerning the first general comment: I recommend authors extend discussions on the biological systems that affect up- and down-regulation of the ncRNAs, that may explain the dysregulation of the expression of ncRNAs,
Answer.
The reviewer is absolutely right, and this is a point that we certainly consider in the beginning. We have not inserted a new paragraph in the introduction to relate about the possible upstream regulations of lncRNA by transcription factors, and their regulation by other epigenetic mechanism. However, lncRNAs appear to be regulated in a similar way as coding genes, using TF binding sites, chromatin marks and histone modifications. To our knowledge, however, nothing specific has been published concerning upstream lncRNA regulations and NDD.
The second key comment is : Authors might better pay more attention and focus on the studies that concerned with the comparison of the expression level of ncRNAs of patients and healthy donors
Answer:
In the text, we provide several examples of levels of ncRNAs impaired in patients with neurodevelopmental disorders. In particular, we reported evidence for LINC00299, GAS5 and SYNGAP1-AS. We now added some information for these lncRNAs, and we also cited works from Wang et al. (https://doi.org/10.1038/tp.2015.144) and Velmeshev et al. (https://doi.org/10.1186/2040-2392-4-32) that were missing from the previous version.
The third key comment is : are there any reports that indicated mutations in some specific lncRNAs in families of ID patients ? That may indicate the direct causative role of the lncRNAs on the pathogenesis of the ID.
Answer
To our knowledge, there are no reports of single nucleotide variants in ncRNAs in patients with ID; however, the disruption of lnc-NR2F1 and LINC00299 has been directly associated with ID, supporting a causative role for lncRNAs in the onset of the disorder. This was indicated in the text and in Table 1 (“lncRNA directly involved in intellectual disabilities”).